# Online Convex Optimization
# with Continuous Switching Constraint

**Guanghui Wang**[1], **Yuanyu Wan**[1,2], **Tianbao Yang**[3], **Lijun Zhang**[1,2,*]

[1]National Key Laboratory for Novel Software Technology, Nanjing University, Nanjing, China
[2]Peng Cheng Laboratory, Shenzhen, Guangdong, China
[3]Department of Computer Science, The University of Iowa, Iowa City, USA
{wanggh,wanyy,zhanglj}@lamda.nju.edu.cn, tianbao-yang@uiowa.edu

## Abstract

In many sequential decision making applications, the change of decision would bring an additional cost, such as the wear-and-tear cost associated with changing server status. To control the switching cost, we introduce the problem of online convex optimization with continuous switching constraint, where the goal is to achieve a small regret given a budget on the *overall* switching cost. We first investigate the hardness of the problem, and provide a lower bound of order $\Omega(\sqrt{T})$ when the switching cost budget $S = \Omega(\sqrt{T})$, and $\Omega(\min\{T/S, T\})$ when $S = O(\sqrt{T})$, where $T$ is the time horizon. The essential idea is to carefully design an adaptive adversary, who can adjust the loss function according to the cumulative switching cost of the player incurred so far based on the orthogonal technique. We then develop a simple gradient-based algorithm which enjoys the minimax optimal regret bound. Finally, we show that, for strongly convex functions, the regret bound can be improved to $O(\log T)$ for $S = \Omega(\log T)$, and $O(\min\{T/\exp(S) + S, T\})$ for $S = O(\log T)$.

## 1 Introduction

Online convex optimization (OCO) is a fundamental framework for studying sequential decision making problems (Shalev-Shwartz, 2011). Its protocol can be seen as a game between a player and an adversary: In each round $t$, firstly, the player selects an action $\mathbf{w}_t$ from a convex set $\mathcal{D} \subseteq \mathbb{R}^d$. After submitting the answer, a loss function $f_t : \mathcal{D} \mapsto \mathbb{R}$ is revealed, and the player suffers a loss $f_t(\mathbf{w}_t)$. The goal is to minimize the regret:

$$R = \sum_{t=1}^{T} f_t(\mathbf{w}_t) - \min_{\mathbf{w} \in \mathcal{D}} \sum_{t=1}^{T} f_t(\mathbf{w}), \tag{1}$$

which is the difference between the cumulative loss of the player and that of the best action in hindsight.

Over the past decades, the problem of OCO has been extensively studied, yielding various algorithms and theoretical guarantees (Hazan, 2016; Orabona, 2019). However, most of the existing approaches allow the player to switch her action *freely* during the learning process. As a result, these methods become unsuitable for many real-life scenarios, such as the online shortest paths problem (Koolen et al., 2010), and portfolio management (Dekel et al., 2014; Vittori et al., 2020), where the switching of actions brings extra cost, and the budget for the overall switching cost is strictly

---

[*]Lijun Zhang is the corresponding author.

35th Conference on Neural Information Processing Systems (NeurIPS 2021).

constrained. To address this problem, recent advances in OCO introduced the switching-constrained problem (Altschuler & Talwar, 2018; Chen et al., 2020), where a hard constraint is imposed to the *number* of the player's action shifts, i.e.,

$$\sum_{t=2}^{T}\{\mathbf{w}_t \neq \mathbf{w}_{t-1}\} \leq K, \tag{2}$$

and the goal is to minimize regret under a fixed budget $K$. For this problem, Chen et al. (2020) have shown that, given any $K$, we could precisely control the the overall switching cost in (2), while achieving a minimax regret bound of order $\Theta(T/\sqrt{K})$.

One limitation of (2) is that it treats different amounts of changes between $\mathbf{w}_{t-1}$ and $\mathbf{w}_t$ equally, since the *binary* function is used as the penalty for action shifts. However, as observed by many practical applications, e.g., thermal management (Zanini et al., 2010), video streaming (Joseph & de Veciana, 2012) and multi-timescale control (Goel et al., 2017), the price paid for large and small action changes are not the same. Specifically, for these scenarios, the switching cost between two consecutive rounds is typically characterized by a $\ell_2$-norm function, i.e., $\|\mathbf{w}_t - \mathbf{w}_{t-1}\|$. Motivated by this observation, in this paper, we introduce a novel OCO setting, named OCO with continuous switching constraint (OCO-CSC), where the player needs to choose actions under a hard constraint on the *overall $\ell_2$-norm switching cost*, i.e.,

$$\sum_{t=2}^{T}\|\mathbf{w}_t - \mathbf{w}_{t-1}\| \leq S, \tag{3}$$

where $S$ is a budget given by the environment. The main advantage of OCO-CSC is that, equipped with (3), we could have a more delicate control on the overall switching cost compared to the binary constraint in (2).

For the proposed problem, we firstly observe that, an $O(T/\sqrt{S})$ regret bound can be achieved by using the method proposed for the switching-constrained OCO (Chen et al., 2020) under a proper configuration of $K$. However, this bound is not tight, since there is a large gap from the lower bound established in this paper. Specifically, we provide a lower bound of order $\Omega(\sqrt{T})$ when $S = \Omega(\sqrt{T})$, and $\Omega(\min\{\frac{T}{S}, T\})$ when $S = O(\sqrt{T})$. Our basic framework for constructing the lower bound follows the classical linear game (Abernethy et al., 2008), while we adopt a novel continuous-constraint-related dynamic policy for the adversary, which allows it to adaptively change the loss function according to the player's cumulative switching costs. Furthermore, we prove that the classical online gradient descent (OGD) with an appropriately chosen step size is able to obtain the matching upper bound. These results demonstrate that there is a *phase transition* phenomenon between large and small switching budget regimes, which is in sharp contrast to the switching-constrained setting, where the minimax bound always decreases with $\Theta(1/\sqrt{K})$. Finally, we propose a variant of OGD for $\lambda$-strongly convex functions, which can achieve an $O(\log T)$ regret bound when $S = \Omega(\log T)$, and an $O(T/\exp(S) + S)$ regret bound when $S = O(\log T)$.

## 2 Related Work

In this section, we briefly review related work on online convex optimization.

### 2.1 Classical OCO

The framework of OCO is established by the seminal work of Zinkevich (2003). For general convex functions, Zinkevich (2003) shows that online gradient descent (OGD) with step size on the order of $O(1/\sqrt{t})$ enjoys an $O(\sqrt{T})$ regret bound. For $\lambda$-strongly convex functions, Hazan et al. (2007) prove that OGD with step size of order $O(1/[\lambda t])$ achieves an $O(\log T)$ regret bound. Both bounds have been proved to be minimax optimal (Abernethy et al., 2008). For exponentially concave functions, the state-of-the-art algorithm is online Newton step (Hazan et al., 2007), which enjoys an $O(d \log T)$ regret bound, where $d$ is the dimensionality.

## 2.2 Switching-constrained OCO

One related line of research is the switching-constrained setting, where the player is only allowed to change her action no more than $K$ times. This setting has been studied in various online learning scenarios, such as prediction with expert advice (Altschuler & Talwar, 2018) and bandits problems (Simchi-Levi & Xu, 2019; Dong et al., 2020; Ruan et al., 2021). In this paper, we focus on online convex optimization. Jaghargh et al. (2019) firstly consider this problem, and develop a novel online algorithm based on the Poison Process, which can achieve an expected regret of order $O(T^{3/2}/\mathrm{E}[K])$ for any given expected switching budget $\mathrm{E}[K]$. Therefore, the regret will become sublinear for $\mathrm{E}[K] = o(\sqrt{T})$. Later, Chen et al. (2020) propose a variant of the classical OGD based on the mini-batch approach. Specifically, the algorithm averagely divides the time horizon into $K$ intervals, and only update the decision at the end of each interval based on OGD. The show that the simple algorithm enjoys an $O(T/\sqrt{K})$ regret bound for any given budget $K$. They also prove that this result is minimax optimal by establishing a matching $\Omega(T/\sqrt{K})$ lower bound. We note that, when the action set is bounded (i.e., $\max_{\mathbf{w}_1,\mathbf{w}_2 \in \mathcal{D}} \|\mathbf{w}_1 - \mathbf{w}_2\| \leq D$), since

$$\sum_{t=2}^{T} \|\mathbf{w}_t - \mathbf{w}_{t-1}\| \leq DK,$$

we could set $K = \lfloor S/D \rfloor$ to satisfy (3) and immediately obtain an $O(T/\sqrt{S})$ regret for OCO-CSC, but there is still a large gap from the lower bound we provide in this paper. Very recently, a concurrent work (Sherman & Koren, 2021) considered switching-constrained OCO under oblivious adversary setting, and derived $\Theta(T/S)$ bounds. However, we note that: (i) lower bound for the switching-constraint OCO does not translate to a lower bound for our continuous switching cost setting; and (ii) their upper bound for oblivious only holds in expectation, and has an undesirable $\sqrt{d}$ factor.

## 2.3 OCO with Ramp Constraints

Another related setting is OCO with ramp constraints, which is studied by Badiei et al. (2015). In this setting, at each round, the player must choose an action satisfying the following inequality:

$$|w_{t,i} - w_{t-1,i}| \leq X_i, \tag{4}$$

where $w_{t,i}$ denotes the $i$-th dimension of $\mathbf{w}_t$, and $X_i$ is a constant factor. The constraint in (4) limits the player's action switching in a *per-round* and *per-dimension* level. This is very different from the constraint we proposed in (3), which mainly focus on the *long-term* and *overall* switching cost. Moreover, we note that, Badiei et al. (2015) assume the player could get access to a sequence of *future* loss functions before choosing $\mathbf{w}_t$, while in this paper we follow the classical OCO framework in which the player can only make use of the historical data.

## 2.4 OCO with Long-term Constraints

Our proposed problem is also related to OCO with long-term constraints (Mahdavi et al., 2012; Jenatton et al., 2016; Yu et al., 2017), where the action set is written as $m$ convex constraints, i.e.,

$$\mathcal{D} = \{\mathbf{w} \in \mathbb{R}^d : g_i(\mathbf{w}) \leq 0, i \in [m]\}, \tag{5}$$

and we only require these constraints to be satisfied in the long term, i.e., $\sum_{t=1}^{T} g_i(\mathbf{w}_t) \leq 0, i \in [m]$. The goal is to minimize regret while keeping $\sum_{t=1}^{T} g_i(\mathbf{w}_t)$ small. We note that, in this setting, the action set is *expressed* by the constraint, which is in contrast to OCO-CSC, where the constraint and the decision set $\mathcal{D}$ are independent. Moreover, the constraint in OCO-CSC is time-variant and decided by the historical decisions, while the constraint in (5) is static (or stochastic, considered by Yu et al., 2017). Recently, several work start to investigate OCO with long-term and time-variant constraints, but this task is proved to be impossible in general (Mannor et al., 2009). Therefore, existing studies have to consider more restricted settings, such as weaker definitions of regret (Neely & Yu, 2017; Liakopoulos et al., 2019; Yi et al., 2020; Valls et al., 2020).

## 2.5 Smoothed OCO

The problem of smoothed OCO is originally proposed in the dynamic right-sizing for power-proportional data centers (Lin et al., 2012b), and has received great research interests during the past decade (Lin et al., 2012a; Bansal et al., 2015; Antoniadis & Schewior, 2017; Chen et al., 2018; Goel et al., 2019; Goel & Wierman, 2019; Zhang et al., 2021a). In smoothed OCO, at each round, the learner will incur a *hitting* cost $f_t(\cdot)$ as well as a *switching* cost $\|\mathbf{w}_t - \mathbf{w}_{t-1}\|$, and the goal is to minimize dynamic regret (Zinkevich, 2003) or competitive ratio (Borodin & El-Yaniv, 2005) with respect to $f_t(\mathbf{w}_t) + \|\mathbf{w}_t - \mathbf{w}_{t-1}\|$. This setting is closely related to but different from OCO-CSC, where the goal is to minimize regret with respect to $f_t(\cdot)$, and the overall switching cost is limited by a given budget. Additionally, we note that, similar to Badiei et al. (2015), studies for the smoothed OCO typically assume the player could see $f_t(\cdot)$ or sometimes a window of future loss functions (Chen et al., 2015, 2016; Li et al., 2018) before choosing $\mathbf{w}_t$. By contrast, in OCO-CSC the player can not obtain these additional information.

## 3 Main Results

In this section, we present the algorithms and theoretical guarantees for OCO-CSC. Before proceeding to the details, following previous work, we introduce some standard definitions (Boyd & Vandenberghe, 2004) and assumptions (Abernethy et al., 2008).

**Defination 1** *A function $f : \mathcal{D} \mapsto \mathbb{R}$ is convex if $\forall \mathbf{w}_1, \mathbf{w}_2 \in \mathcal{D}$,*

$$f(\mathbf{w}_1) \geq f(\mathbf{w}_2) + \nabla f(\mathbf{w}_2)^\top (\mathbf{w}_1 - \mathbf{w}_2). \tag{6}$$

**Defination 2** *A function $f : \mathcal{D} \mapsto \mathbb{R}$ is $\lambda$-strongly convex if $\forall \mathbf{w}_1, \mathbf{w}_2 \in \mathcal{D}$,*

$$f(\mathbf{w}_1) \geq f(\mathbf{w}_2) + \nabla f(\mathbf{w}_2)^\top (\mathbf{w}_1 - \mathbf{w}_2) + \frac{\lambda}{2}\|\mathbf{w}_1 - \mathbf{w}_2\|^2. \tag{7}$$

**Assumption 1** *$\mathcal{D}$ is a $d$-dimensional ball of radius $\frac{D}{2}$, i.e., $\mathcal{D} = \{\mathbf{w}|\mathbf{w} \in \mathbb{R}^d, \|\mathbf{w}\| \leq \frac{D}{2}\}$.*

**Assumption 2** *The gradients of all the online functions are bounded by $G$, i.e.,*

$$\max_{\mathbf{w} \in \mathcal{D}} \|\nabla f_t(\mathbf{w})\| \leq G, \ \forall t \in [T]. \tag{8}$$

### 3.1 Lower Bound for Convex Functions

---
**Algorithm 1** Adversary's Policy
---
1: $\tau = 1, l_\tau = 1$
2: Observe the player's action $\mathbf{w}_1$
3: Choose $\mathbf{m}_1$ such that $\mathbf{m}_1^\top \mathbf{w}_1 = 0$
4: **for** $t = 2$ **to** $T$ **do**
5:     Observe the player's action $\mathbf{w}_t$
6:     **if** $\sum_{j=l_\tau+1}^{t} \|\mathbf{w}_j - \mathbf{w}_{j-1}\| \leq \frac{c}{S}$ **then**
7:         Choose $\mathbf{m}_t = \mathbf{m}_{t-1}$
8:     **else**
9:         Choose $\mathbf{m}_t$ such that $\mathbf{m}_t^\top \mathbf{w}_t \geq 0$ and $\mathbf{m}_t^\top \left(\sum_{j=1}^{t-1} \mathbf{m}_j\right) \geq 0$.
10:         Set $\tau = \tau + 1, l_\tau = t$
11:     **end if**
12: **end for**
---

We first describe the adversary's policy for obtaining the lower bound. Following previous work Abernethy et al. (2008), our proposed policy is based on the *linear game*, i.e., in each round, the adversary chooses from a set of bounded linear functions:

$$F = \{f(\cdot) : \mathcal{D} \mapsto \mathbb{R}|f(\mathbf{w}) = \mathbf{m}^\top \mathbf{w}, \|\mathbf{m}\| = G\},$$

which is a subset of convex functions satisfying Assumptions 1 and 2. For this setting, the regret can be written as

$$\sum_{t=1}^{T} f_t(\mathbf{w}_t) - \min_{\mathbf{w} \in \mathcal{D}} \sum_{t=1}^{T} f_t(\mathbf{w}) = \sum_{t=1}^{T} \mathbf{m}_t^\top \mathbf{w}_t - \min_{\mathbf{w} \in \mathcal{D}} \left( \sum_{t=1}^{T} \mathbf{m}_t \right)^\top \mathbf{w} = \sum_{t=1}^{T} \mathbf{m}_t^\top \mathbf{w}_t + \frac{D}{2} \left\| \sum_{t=1}^{T} \mathbf{m}_t \right\|, \tag{9}$$

where the third equality is because the minimum is only obtained when

$$\mathbf{w} = -D \frac{\sum_{t=1}^{T} \mathbf{m}_t}{2 \| \sum_{t=1}^{T} \mathbf{m}_t \|}.$$

According to (9), to get a tight lower bound for $R$, we have to make both $\sum_{t=1}^{T} \mathbf{m}_t^\top \mathbf{w}_t$ and $\| \sum_{t=1}^{T} \mathbf{m}_t \|$ as large as possible. One classical way to achieve this goal is through the *orthogonal technique* (Abernethy et al., 2008; Chen et al., 2020), that is, in round $t$, the adversary chooses $\mathbf{m}_t$ such that $\mathbf{m}_t^\top \mathbf{w}_t \geq 0$ and $\mathbf{m}_t^\top (\sum_{i=1}^{t-1} \mathbf{m}_i) \geq 0$. Note that such a $\mathbf{m}_t$ can always be found for $d \geq 2$. For this technique, it can be easily shown that $\sum_{t=1}^{T} \mathbf{m}_t^\top \mathbf{w}_t \geq 0$, while $\| \sum_{t=1}^{T} \mathbf{m}_t \| \geq G\sqrt{T}$, which implies an $\Omega(DG\sqrt{T})$ lower bound.

The above policy does not take the constraint on the player's action shifts into account. In the following, we show that, by designing a more adaptive adversary which automatically adjusts its action based on the player's historical switching costs, we can obtain a tighter lower bound when $S$ is small. The details is summarized in Algorithm 1. Specifically, in the first round, after observing the player's action $\mathbf{w}_1$, the adversary just simply chooses $f_1(\mathbf{w}) = \mathbf{m}_1^\top \mathbf{w}$ such that $\mathbf{m}_1^\top \mathbf{w}_1 = 0$ (Step 3). For round $t \geq 2$, the adversary divides the time horizon into several epochs. Let the number of epochs be $N$. For each round $t$ in epoch $\tau \in [N]$, after obtaining $\mathbf{w}_t$, the adversary checks if the cumulative switching cost of the player inside epoch $\tau$ exceeds a threshold (Step 6). To be more specific, the adversary will check if

$$\sum_{j=l_\tau+1}^{t} \| \mathbf{w}_j - \mathbf{w}_{j-1} \| \leq \frac{c}{S},$$

where $l_\tau$ is the start point of epoch $\tau$, $\ell_1 = 1$, and $c > 0$ is a constant factor. If the inequality holds, then the adversary will keep the action unchanged (Step 7); otherwise, the adversary will find a new $\mathbf{m}_t$ based on the orthogonal technique, i.e., find $\mathbf{m}_t$ such that $\mathbf{m}_t^\top \mathbf{w}_t \geq 0$ and $\mathbf{m}_t^\top M_{t-1} \geq 0$, where $M_{t-1} = \sum_{j=1}^{t-1} \mathbf{m}_j$, and then start a new epoch (Steps 9-10).

The essential idea behind the above policy is that the adversary adaptively divides $T$ iterations into $N$ epochs, such that for each epoch $\tau \in [N]$, the cumulative switching cost inside of $\tau$ is upper bounded by

$$\sum_{j=l_\tau+1}^{l_{\tau+1}-1} \| \mathbf{w}_j - \mathbf{w}_{j-1} \| \leq \frac{c}{S},$$

and for each epoch $\tau \in [N-1]$,

$$\sum_{j=l_\tau+1}^{l_{\tau+1}} \| \mathbf{w}_j - \mathbf{w}_{j-1} \| > \frac{c}{S}.$$

The above two inequalities help us obtain novel lower bounds for the two terms at the R.H.S. of (9) respectively which depend on $S$. Specifically, we prove the following two lemmas.

**Lemma 1** *We have*

$$\sum_{t=1}^{T} \mathbf{m}_t^\top \mathbf{w}_t \geq -\frac{cGT}{S}.$$

**Lemma 2** *We have*

$$\left\| \sum_{t=1}^{T} \mathbf{m}_t \right\| \geq G \frac{T\sqrt{c}}{\sqrt{S^2 + c}}.$$

By appropriately tuning the parameter $c$, we finally prove the following lower bound.

**Theorem 1** *For any online algorithm, under any given switching cost budget $S$, Algorithm 1 can generate a series of loss functions $f_1(\cdot), \ldots, f_T(\cdot)$ satisfying Assumptions 1 and 2, such that*

$$R \geq \begin{cases} 0.5DG\sqrt{T}, & S \in [D\sqrt{T}, DT] \\ 0.05DG\frac{DT}{S}, & S \in [D, D\sqrt{T}) \\ 0.05DGT, & S \in [0, D). \end{cases}$$

**Remark 1** The above theorem implies that, when $S \leq D$, the lower bound for OCO-CSC is linear with respect to $T$; When $S \in [D, D\sqrt{T})$, it's possible to achieve sublinear results, and the lower bound decreases with $T/S$; for sufficiently large $S$, i.e., when $S = \Omega(D\sqrt{T})$, the lower bound is $\Omega(DG\sqrt{T})$, which matches the lower bound for the general OCO problem (Abernethy et al., 2008). Note that in this case the lower bound will not further improve as $S$ increases, which is very different from the switching-constrained setting, where the lower bound is $\Omega(T/\sqrt{K})$, which means that increasing the budget $K$ is always beneficial. Finally, we note that, since

$$\frac{1}{D}\sum_{t=2}^{T} \|\mathbf{w}_t - \mathbf{w}_{t-1}\| \leq \sum_{t=2}^{T}\{\mathbf{w}_t \neq \mathbf{w}_{t-1}\},$$

lower bound for the binary switching cost setting (Altschuler & Talwar, 2018; Chen et al., 2020) does not translate to lower bound for our continuous switching cost setting.

**Remark 2** We summarize our main ideas for proving Theorem 1 as follows. For the proposed adversary, we firstly show that (Lemma 1, Eq. (26)), inside of each batch, there exists a negative error term caused by the possible *non-orthogonality* between the player's decisions and the adversary's choice. To maximize this error term and further the lower bound, the adversary should change its choice as many times as possible (thus require a *small* threshold); On the other hand, Lemma 2 (Eq. (30)) indicates that, the lower bound is tighter when the number of batches is small (thus require a *large* threshold). Based on the two lemmas, in the proof of Theorem 1, we show that the final lower bound is a function of the threshold. Very luckily, we find that the *optimal* lower bound can be derived by choosing a threshold that maximizes the function.

### 3.2 Upper Bounds

In this section, we provide the algorithm for obtaining the upper bound. Before introducing our method, we note that, as mentioned in Section 2.2, the mini-batch OGD algorithm proposed by Chen et al. (2020) enjoys an $O(T/\sqrt{S})$ regret bound for OCO-CSC, which is suboptimal based on the lower bound we constructed at the last section. In the following, we show that, perhaps a bit surprisingly, the classical online gradient descent with an appropriately chosen step size is sufficient for obtaining the matching upper bound. Specifically, in round $t$, we update $\mathbf{w}_t$ by

$$\mathbf{w}_{t+1} = \Pi_{\mathcal{D}}\left[\mathbf{w}_t - \eta \nabla f_t(\mathbf{w}_t)\right], \tag{10}$$

where $\Pi_{\mathcal{D}}[\mathbf{p}]$ denotes projecting $\mathbf{p}$ into $\mathcal{D}$, i.e.,

$$\Pi_{\mathcal{D}}[\mathbf{p}] = \underset{\mathbf{w} \in \mathcal{D}}{\operatorname{argmin}}(\mathbf{w} - \mathbf{p})^{\top}(\mathbf{w} - \mathbf{p}).$$

For this algorithm, we prove the following theoretical guarantee.

**Theorem 2** *Suppose Assumptions 1 and 2 hold, and all loss functions are convex. Then, under any given switching cost budget $S$, OGD with step size*

$$\eta = \begin{cases} \frac{D}{G\sqrt{T}}, & S \in [D\sqrt{T}, DT] \\ \frac{S}{GT}, & S \in [0, D\sqrt{T}) \end{cases} \tag{11}$$

*satisfies (3), and achieves the following regret:*

$$R \leq \begin{cases} DG\sqrt{T}, & S \in [D\sqrt{T}, DT] \\ DG\frac{DT}{S}, & S \in [D, D\sqrt{T}) \\ DGT, & S \in [0, D). \end{cases}$$

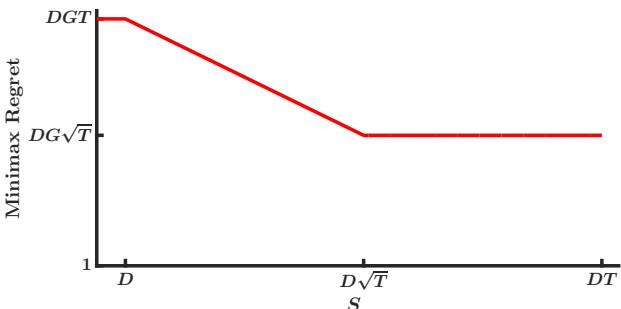

Figure 1: Minimax regret of OCO-CSC. Axies are plotted in log-log scale.

**Remark 3** Theorems 1 and 2 show that our proposed algorithm enjoys an $O(GD\sqrt{T})$ minimax regret bound for $S = \Omega(D\sqrt{T})$, and $O(DG \min\{DT/S, T\})$ regret bound for $S = O(D\sqrt{T})$. We illustrate the relationship between the minimax regret and $S$ in Figure 1. Finally, we note that the algorithm requires the paramter $T$ as input, but it could be easily extended to an any-time algorithm by using the doubling trick Shalev-Shwartz (2011).

Although the analysis above implies that the theoretical guarantee for OCO-CSC is unimproveable in general, in the following, we show that tighter bounds is still achievable when the loss functions are strongly convex. Specifically, when there are no constraints on the switching cost, the state-of-the-art algorithm is OGD with a time variant step size $\eta_t = 1/[\lambda t]$, which enjoys an $O(\log T)$ regret bound. For the OCO-CSC setting, in order to control the overall switching cost, we propose to add a tuning parameter at the denominator of the step size. To be more specific, in round $t$, we update $\mathbf{w}_t$ by

$$\mathbf{w}_{t+1} = \Pi_{\mathcal{D}} \left[ \mathbf{w}_t - \eta_t \nabla f_t(\mathbf{w}_t) \right], \tag{12}$$

where $\eta_t = \frac{1}{\lambda(t+c)}$, and $c > 0$ is a constant factor. By configuring $c$ properly, we can obtain the following regret bound.

**Theorem 3** *Suppose Assumptions 1 and 2 hold, and all loss functions are $\lambda$-strongly convex. Then, under any given switching cost budget $S$, the algorithm in (12) with*

$$c = \begin{cases} 0, & S \in [\frac{2G}{\lambda} \log(T+1), DT] \\ \frac{T}{\exp(\frac{\lambda}{2G}S)-1} - 1, & S \in [0, \frac{2G}{\lambda} \log(T+1)) \end{cases} \tag{13}$$

*satisfies (3), and achieves $R \le \lambda D^2 + \frac{2G^2}{\lambda} \log(T+1)$. for $S \in [\frac{2G}{\lambda} \log(T+1), DT]$, and*

$$R \le \min \left\{ \frac{\lambda T D^2}{\exp(\frac{\lambda}{2G}S) - 1} + GS, DGT \right\}$$

*for $S \in [0, \frac{2G}{\lambda} \log(T+1))$.*

**Remark 4** Theorem 3 implies that, when $S \ge \frac{2G}{\lambda} \log(T+1)$, the proposed algorithm enjoys an $O(\log T)$ optimal regret bound; for $S \le \frac{2G}{\lambda} \log(T+1)$, the proposed algorithm achieves an $O(T/\exp(S) + S)$ regret bound. To obtain a sublinear regret bound, consider $S = \frac{2G}{\lambda} \log(T^\alpha + 1)$. In this case, we have

$$R \le \lambda D^2 T^{1-\alpha} + \frac{2G^2}{\lambda} \log(T^\alpha + 1),$$

which is sublinar for $\alpha \in (0, 1]$.

## 4 Theoretical Analysis

In this section, we present the proofs for the main conclusions.

## 4.1 Proof of Theorem 1

When $S \geq D\sqrt{T}$, the lower bound can be directly obtained by using the minimax linear game provided by (Abernethy et al., 2008). When $S \in [D, D\sqrt{T})$, by the definition of regret, we have

$$R = \underbrace{\sum_{t=1}^{T} \mathbf{m}_t^\top \mathbf{w}_t}_{a_1} - \underbrace{\min_{\mathbf{w} \in \mathcal{D}} \left( \sum_{t=1}^{T} \mathbf{m}_t \right)^\top \mathbf{w}}_{a_2}.$$

Based on Lemmas 1 and 2, we get

$$R = a_1 + a_2 \geq G \frac{0.5DT\sqrt{c}}{\sqrt{S^2 + c}} - \frac{cGT}{S}. \tag{14}$$

Let $c = c'D^2$, and we have

$$R \geq G \frac{0.5DT\sqrt{c'D^2}}{\sqrt{S^2 + c'D^2}} - \frac{c'D^2GT}{S} = GD^2T \left( \frac{0.5\sqrt{c'}}{\sqrt{S^2 + c'D^2}} - \frac{c'}{S} \right) \geq GD^2 \frac{T}{S} \left( \frac{0.5\sqrt{c'}}{\sqrt{1 + c'}} - c' \right).$$

where the second inequality is due to $D \leq S$. Note that the R.H.S. of the above inequality is a function of $c'$. To maximize the lower bound, we should solve the following convex problem:

$$\underset{x > 0}{\operatorname{argmax}} \frac{0.5\sqrt{x}}{\sqrt{1 + x}} - x,$$

which is equivalent to finding the solution of the following equation:

$$16x^4 + 32x^3 + 49x^2 + 15x - 1 = 0.$$

it can be easily shown that the optimal solution $x_* \approx 0.056$. Thus, by setting $c' = 0.056$, we get

$$R \geq 0.05GD^2 \frac{T}{S}. \tag{15}$$

For $S \in (0, D]$, based on (14) and setting $c = 0.056S^2$, we have

$$R \geq GT \left( \frac{0.5D\sqrt{0.056S^2}}{\sqrt{S^2 + 0.056S^2}} - \frac{0.056S^2}{S} \right) \geq GDT \left( \frac{0.028}{\sqrt{1.056}} - 0.056 \right) \geq 0.05GDT.$$

where the second inequality is because $S \leq D$.

## 4.2 Proof of Theorem 2

We first prove that by setting $\eta$ as in (11), the constraint in (3) always holds. Let $\mathbf{w}_t' = \mathbf{w}_{t-1} - \eta \nabla f_{t-1}(\mathbf{w}_{t-1})$. We have

$$\sum_{t=2}^{T} \|\mathbf{w}_t - \mathbf{w}_{t-1}\| \leq \sum_{t=2}^{T} \|\mathbf{w}_t' - \mathbf{w}_{t-1}\| \overset{(10)}{=} \sum_{t=2}^{T} \|\eta \nabla f_{t-1}(\mathbf{w}_{t-1})\| \overset{(8)}{\leq} \eta GT, \tag{16}$$

where the first inequality is based the following lemma, which describes the non-expansion property of the projection.

**Lemma 3** (McMahan & Streeter, 2010) *For the projection operation, we have $\forall \mathbf{w}_1, \mathbf{w}_2 \in \mathbb{R}^d$,*

$$\|\Pi_{\mathcal{D}}(\mathbf{w}_1) - \Pi_{\mathcal{D}}(\mathbf{w}_2)\| \leq \|\mathbf{w}_1 - \mathbf{w}_2\|.$$

Based on (16), for $S \in [D\sqrt{T}, DT]$, we have

$$\sum_{t=2}^{T} \|\mathbf{w}_t - \mathbf{w}_{t-1}\| \leq \eta GT = D\sqrt{T} \leq S.$$

For $S \in [0, D\sqrt{T})$, we have

$$\sum_{t=2}^{T} \|\mathbf{w}_t - \mathbf{w}_{t-1}\| \leq \eta G T = \frac{S}{GT} G T = S.$$

Next, we turn to upper bound the regret. Let $\mathbf{w}_* = \operatorname{argmin}_{\mathbf{w} \in \mathcal{D}} \sum_{t=1}^{T} f_t(\mathbf{w})$. Based on the classical analysis of OGD (Hazan, 2016), we have

$$
\begin{aligned}
\|\mathbf{w}_t - \mathbf{w}_*\|^2 \leq \|\mathbf{w}_t' - \mathbf{w}_*\|^2 =& \|\mathbf{w}_{t-1} - \eta \nabla f_{t-1}(\mathbf{w}_{t-1}) - \mathbf{w}_*\|^2 \\
=& \|\mathbf{w}_{t-1} - \mathbf{w}_*\|^2 + \eta^2 \|\nabla f_{t-1}(\mathbf{w}_{t-1})\|^2 \\
& - 2\eta (\mathbf{w}_{t-1} - \mathbf{w}_*)^\top \nabla f_{t-1}(\mathbf{w}_{t-1}).
\end{aligned}
$$

Thus

$$(\mathbf{w}_t - \mathbf{w}_*)^\top \nabla f_t(\mathbf{w}_t) \leq \frac{\|\mathbf{w}_t - \mathbf{w}_*\|^2 - \|\mathbf{w}_{t+1} - \mathbf{w}_*\|^2}{2\eta} + \frac{\eta}{2} \|\nabla f_t(\mathbf{w}_t)\|^2.$$

Based on the convexity of the loss functions and summing the above inequality up from 1 to $T$, we get

$$R \leq \sum_{t=1}^{T} \frac{\|\mathbf{w}_t - \mathbf{w}_*\|^2 - \|\mathbf{w}_{t+1} - \mathbf{w}_*\|^2}{2\eta} + \frac{\eta}{2} \sum_{t=1}^{T} \|\nabla f_t(\mathbf{w}_t)\|^2 \leq \frac{D^2}{2\eta} + \frac{\eta G^2 T}{2}. \qquad (17)$$

Thus, for $S \in [D\sqrt{T}, DT]$, we have

$$R \leq \frac{D^2}{2\eta} + \frac{\eta G^2 T}{2} = DG\sqrt{T}. \qquad (18)$$

For $S \in [0, D\sqrt{T})$,

$$R \leq \frac{D^2}{2\eta} + \frac{\eta G^2 T}{2} = DG \left( \frac{DT}{2S} + \frac{S}{2D} \right) \leq DG \left( \frac{DT}{2S} + \frac{\sqrt{T}}{2} \right) \leq DG \left( \frac{DT}{2S} + \frac{DT}{2S} \right) = DG \frac{DT}{S}.$$

Finally, note that by Assumptions 1 and 2 and the convexity of the loss functions, we always have $R \leq DGT$.

## 5   Conclusion and Future Work

In this paper, we propose a variant of the classical OCO problem, named OCO with continuous switching constraint, where the player suffers a $\ell_2$-norm switching cost for each action shift, and the overall switching cost is constrained by a given budget $S$. We first propose an adaptive mini-batch policy for the adversary, based on which we prove that the lower bound for this problem is $\Omega(\sqrt{T})$ when $S = \Omega(\sqrt{T})$, and $\Omega(\min\{\frac{T}{S}, T\})$ when $S = O(\sqrt{T})$. Next, we demonstrate that OGD with a proper configuration of the step size achieves the minimax optimal regret bound. Finally, for $\lambda$-strongly convex functions, we develop a variant of OGD, which has a tunable parameter at the denominator, and we show that it enjoys an $O(\log T)$ regret bound when $S = \Omega(\log T)$, and an $O(T/\exp(S) + S)$ regret bound when $S = O(\log T)$.

In the future, we would like to investigate how to extend the proposed constraint setting to other OCO scenarios, such as minimizing adaptive regret (Jun et al., 2017; Zhang et al., 2019), dynamic regret (Zinkevich, 2003; Zhang et al., 2018), and Universal OCO (van Erven & Koolen, 2016; Wang et al., 2019; Zhang et al., 2021b). Moreover, it is also an interesting question to study the switching constraint problem under other distance metrics such as the Bregman divergence.

## Acknowledgments and Disclosure of Funding

Funding in direct support of this work: National Key Research and Development Program of China (2018AAA0101100), NSFC grant 61921006 and 61976112.

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
