$$R \geq GT \left( \frac{0.5 D\sqrt{0.056 S^2}}{\sqrt{S^2 + 0.056 S^2}} - \frac{0.056 S^2}{S} \right) \geq GDT \left( \frac{0.028}{\sqrt{1.056}} - 0.056 \right) \geq 0.05 GDT.$$

where the second inequality is because $S \leq D$.

### 4.2 Proof of Theorem 2

We first prove that by setting $\eta$ as in (11), the constraint in (3) always holds. Let $\mathbf{w}'_t = \mathbf{w}_{t-1} - \eta \nabla f_{t-1}(\mathbf{w}_{t-1})$. We have

$$\sum_{t=2}^{T} \|\mathbf{w}_t - \mathbf{w}_{t-1}\| \leq \sum_{t=2}^{T} \|\mathbf{w}'_t - \mathbf{w}_{t-1}\| \overset{(10)}{=} \sum_{t=2}^{T} \|\eta \nabla f_{t-1}(\mathbf{w}_{t-1})\| \overset{(8)}{\leq} \eta GT, \tag{16}$$

where the first inequality is based the following lemma, which describes the non-expansion property of the projection.

**Lemma 3** (McMahan & Streeter, 2010) *For the projection operation, we have* $\forall \mathbf{w}_1, \mathbf{w}_2 \in \mathbb{R}^d$,

$$\|\Pi_{\mathcal{D}}(\mathbf{w}_1) - \Pi_{\mathcal{D}}(\mathbf{w}_2)\| \leq \|\mathbf{w}_1 - \mathbf{w}_2\|.$$

Based on (16), for $S \in [D\sqrt{T}, DT]$, we have

$$\sum_{t=2}^{T} \|\mathbf{w}_t - \mathbf{w}_{t-1}\| \leq \eta GT = D\sqrt{T} \leq S.$$

For $S \in [0, D\sqrt{T})$, we have

$$\sum_{t=2}^{T} \|\mathbf{w}_t - \mathbf{w}_{t-1}\| \leq \eta GT = \frac{S}{GT}GT = S.$$

Next, we turn to upper bound the regret. Let $\mathbf{w}_* = \arg\min_{\mathbf{w} \in \mathcal{D}} \sum_{t=1}^{T} f_t(\mathbf{w})$. Based on the classical analysis of OGD (Hazan, 2016), we have

$$\|\mathbf{w}_t - \mathbf{w}_*\|^2 \leq \|\mathbf{w}'_t - \mathbf{w}_*\|^2 = \|\mathbf{w}_{t-1} - \eta\nabla f_{t-1}(\mathbf{w}_{t-1}) - \mathbf{w}_*\|^2$$
$$= \|\mathbf{w}_{t-1} - \mathbf{w}_*\|^2 + \eta^2\|\nabla f_{t-1}(\mathbf{w}_{t-1})\|^2$$
$$- 2\eta(\mathbf{w}_{t-1} - \mathbf{w}_*)^\top \nabla f_{t-1}(\mathbf{w}_{t-1}).$$

Thus

$$(\mathbf{w}_t - \mathbf{w}_*)^\top \nabla f_t(\mathbf{w}_t) \leq \frac{\|\mathbf{w}_t - \mathbf{w}_*\|^2 - \|\mathbf{w}_{t+1} - \mathbf{w}_*\|^2}{2\eta} + \frac{\eta}{2}\|\nabla f_t(\mathbf{w}_t)\|^2.$$

Based on the convexity of the loss functions and summing the above inequality up from 1 to $T$, we get

$$R \leq \sum_{t=1}^{T} \frac{\|\mathbf{w}_t - \mathbf{w}_*\|^2 - \|\mathbf{w}_{t+1} - \mathbf{w}_*\|^2}{2\eta} + \frac{\eta}{2} \sum_{t=1}^{T} \|\nabla f_t(\mathbf{w}_t)\|^2 \leq \frac{D^2}{2\eta} + \frac{\eta G^2 T}{2}. \quad (17)$$

Thus, for $S \in [D\sqrt{T}, DT]$, we have

$$R \leq \frac{D^2}{2\eta} + \frac{\eta G^2 T}{2} = DG\sqrt{T}. \quad (18)$$

For $S \in [0, D\sqrt{T})$,

$$R \leq \frac{D^2}{2\eta} + \frac{\eta G^2 T}{2} = DG\left(\frac{DT}{2S} + \frac{S}{2D}\right) \leq DG\left(\frac{DT}{2S} + \frac{\sqrt{T}}{2}\right) \leq DG\left(\frac{DT}{2S} + \frac{DT}{2S}\right) = DG\frac{DT}{S}.$$

Finally, note that by Assumptions 1 and 2 and the convexity of the loss functions, we always have $R \leq DGT$.

## 5   Conclusion and Future Work

In this paper, we propose a variant of the classical OCO problem, named OCO with continuous switching constraint, where the player suffers a $\ell_2$-norm switching cost for each action shift, and the overall switching cost is constrained by a given budget $S$. We first propose an adaptive mini-batch policy for the adversary, based on which we prove that the lower bound for this problem is $\Omega(\sqrt{T})$ when $S = \Omega(\sqrt{T})$, and $\Omega(\min\{\frac{T}{S}, T\})$ when $S = O(\sqrt{T})$. Next, we demonstrate that OGD with a proper configuration of the step size achieves the minimax optimal regret bound. Finally, for $\lambda$-strongly convex functions, we develop a variant of OGD, which has a tunable parameter at the denominator, and we show that it enjoys an $O(\log T)$ regret bound when $S = \Omega(\log T)$, and an $O(T/\exp(S) + S)$ regret bound when $S = O(\log T)$.

In the future, we would like to investigate how to extend the proposed constraint setting to other OCO scenarios, such as minimizing adaptive regret (Jun et al., 2017; Zhang et al., 2019), dynamic regret (Zinkevich, 2003; Zhang et al., 2018), and Universal OCO (van Erven & Koolen, 2016; Wang et al., 2019; Zhang et al., 2021b). Moreover, it is also an interesting question to study the switching constraint problem under other distance metrics such as the Bregman divergence.

## Acknowledgments and Disclosure of Funding

Funding in direct support of this work: National Key Research and Development Program of China (2018AAA0101100), NSFC grant 61921006 and 61976112.

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

# A  Proof of Theorem 3

Let $\mathbf{w}'_t = \mathbf{w}_{t-1} - \eta_{t-1}\nabla f_{t-1}(\mathbf{w}_{t-1})$. We have

$$\sum_{t=2}^{T} \|\mathbf{w}_t - \mathbf{w}_{t-1}\| \leq \sum_{t=2}^{T} \|\mathbf{w}'_t - \mathbf{w}_{t-1}\| \leq \frac{G}{\lambda} \sum_{t=1}^{T} \frac{1}{t+c}. \tag{19}$$

where the first inequality is based on Lemma 3. To further upper bound the theorem, we introduce the following lemma.

**Lemma 4** (Gaillard et al., 2014) *Let $a_0 > 0$ and $a_1, \ldots, a_m \in [0,1]$ be real numbers and let $f : [0, +\infty) \mapsto [0, +\infty)$ be a nonincreasing function. then*

$$\sum_{i=2}^{m} a_i f(a_0 + \cdots + a_{i-1}) \leq f(a_0) + \int_{a_0}^{a_0 + \cdots + a_m} f(x) dx.$$

Based on the lemma above, we have

$$\sum_{t=2}^{T} \|\mathbf{w}_t - \mathbf{w}_{t-1}\| \leq \frac{G}{\lambda} \frac{1}{1+c} + \frac{G}{\lambda} \left( \log(T+c) - \log(1+c) \right)$$

$$\leq \frac{G}{\lambda} \frac{1}{1+c} + \frac{G}{\lambda} \log\left( \frac{T}{1+c} + 1 \right) \leq \frac{2G}{\lambda} \log\left( \frac{T}{1+c} + 1 \right), \tag{20}$$

where the last inequality is because $1/x \leq \log(T/x + 1)$ for any $x \geq 1$ and $T \geq 3$. Thus, for $S \geq \frac{2G}{\lambda} \log(T+1)$, by setting $c = 0$, we have

$$\sum_{t=2}^{T} \|\mathbf{w}_t - \mathbf{w}_{t-1}\| \leq \frac{2G}{\lambda} \log\left( \frac{T}{1+c} + 1 \right) \leq S.$$

When $S \leq \frac{2G}{\lambda} \log(T+1)$, we configure

$$c = \frac{T}{\exp(\frac{\lambda}{2G}S) - 1} - 1 \geq 0,$$

and get

$$\sum_{t=1}^{T} \|\mathbf{w}_t - \mathbf{w}_{t-1}\| \leq S.$$

Next, we consider the regret bound. For $t \in [1, T]$, We have

$$\|\mathbf{w}_{t+1} - \mathbf{w}_*\|^2 \leq \|\mathbf{w}'_{t+1} - \mathbf{w}_*\|^2$$

$$= \|\mathbf{w}_t - \mathbf{w}_*\|^2 - 2\eta_t(\mathbf{w}_t - \mathbf{w}_*)^\top \nabla f_t(\mathbf{w}_t) + \eta_t^2 \|\nabla f_t(\mathbf{w}_t)\|^2, \tag{21}$$

thus

$$(\mathbf{w}_t - \mathbf{w}_*)^\top \nabla f_t(\mathbf{w}_t) \leq \frac{\|\mathbf{w}_t - \mathbf{w}_*\|^2 - \|\mathbf{w}_{t+1} - \mathbf{w}_t\|^2}{2\eta_t} + \frac{\eta_t}{2} \|\nabla f_t(\mathbf{w}_t)\|^2. \tag{22}$$

By the definition of regret and strong convexity, we have

$$R = \sum_{t=1}^{T} f_t(\mathbf{w}_t) - \sum_{t=1}^{T} f_t(\mathbf{w}_*)$$

$$\leq \sum_{t=1}^{T} (\mathbf{w}_t - \mathbf{w}_*)^\top \nabla f_t(\mathbf{w}_t) - \frac{\lambda}{2} \sum_{t=1}^{T} \|\mathbf{w}_t - \mathbf{w}_*\|^2$$

$$\leq \frac{1}{2} \sum_{t=2}^{T} \underbrace{\left( \frac{1}{\eta_t} - \frac{1}{\eta_{t-1}} - \lambda \right)}_{=0} \|\mathbf{w}_t - \mathbf{w}_*\|^2 + \sum_{t=1}^{T} \frac{\eta_t}{2} G^2 + \frac{D^2}{2\eta_1}$$

$$\leq \lambda(c+1)D^2 + \frac{2G^2}{\lambda} \log\left( \frac{T}{1+c} + 1 \right).$$

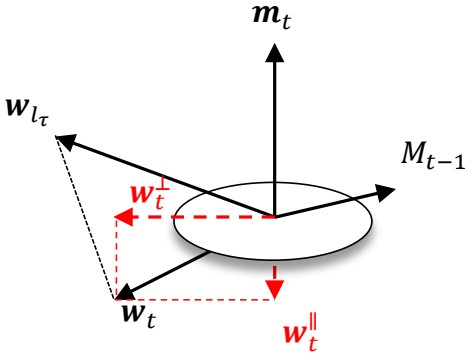

Figure 2: Decomposition of $\mathbf{w}_t$ when $d = 3$.

Thus, for $S \geq \frac{2G}{\lambda} \log(T+1)$, we have $c = 0$, and thus

$$R \leq \lambda D^2 + \frac{2G^2}{\lambda} \log\left(T+1\right).$$

When $S \leq \frac{2G}{\lambda} \log(T+1)$, we have $c = \frac{T}{\exp(\frac{\lambda}{2G}S)-1} - 1$, thus

$$R \leq \frac{\lambda T D^2}{\exp(\frac{\lambda}{2G}S) - 1} + GS.$$

Finally, under Assumptions 1 and 2, we always have $R \leq DGT$.

## B  Proof of Lemma 1

For any epoch $\tau \in [N]$ of length 1, we have

$$\mathbf{m}_{l_\tau}^\top \mathbf{w}_{l_\tau} \geq 0. \tag{23}$$

For any epoch $\tau \in [N]$ whose length is greater than 1, we have $\forall t \in [l_\tau, l_{\tau+1} - 1]$,

$$\|\mathbf{w}_t - \mathbf{w}_{l_\tau}\| \leq \sum_{j=l_\tau+1}^{t} \|\mathbf{w}_j - \mathbf{w}_{j-1}\| \leq \frac{c}{S}, \tag{24}$$

where the first inequality is based on the triangle inequality, and the second inequality is guaranteed by Step 6 of the adversary's policy. Next, we decompose $\mathbf{w}_t$ into two terms: $\mathbf{w}_t = \mathbf{w}_t^\parallel + \mathbf{w}_t^\perp$, where $\mathbf{w}_t^\parallel$ is the component parallel to $\mathbf{m}_t$, and $\mathbf{w}_t^\perp$ the component parallel to the $(d-1)$-normal-hyperplane of $\mathbf{m}_t$. We illustrate the decomposition for $d = 3$ in Figure 2. Based on the decomposition, we have

$$\mathbf{m}_t^\top \mathbf{w}_t = \mathbf{m}_t^\top (\mathbf{w}_t^\perp + \mathbf{w}_t^\parallel) = \mathbf{m}_t^\top \mathbf{w}_t^\parallel = -G\|\mathbf{w}_t^\parallel\| \geq -G\|\mathbf{w}_t - \mathbf{w}_{l_\tau}\| \geq -\frac{cG}{S}, \tag{25}$$

where the first inequality is based on the triangle inequality, and the fact that $\mathbf{w}_{l_\tau}$ is always above the $(d-1)$-normal-hyperplane. The second inequality is derived from (24). Combining (23) and (25), we know that for $t \in [T]$, $\mathbf{m}_t^\top \mathbf{w}_t \geq -\frac{cG}{S}$, and thus

$$\sum_{t=1}^{T} \mathbf{m}_t^\top \mathbf{w}_t \geq -\frac{cGT}{S}. \tag{26}$$

## C  Proof of Lemma 2

Based on Step 6 of the adversary's policy, we know that, for epoch $\tau \in [N-1]$,

$$\sum_{j=l_\tau+1}^{l_{\tau+1}} \|\mathbf{w}_j - \mathbf{w}_{j-1}\| > \frac{c}{S}, \tag{27}$$

since otherwise the adversary will not start a new epoch at $\ell_{\tau+1}$ (note that the cumulative switching cost at the last epoch does not have this lower bound). Thus, the total switching cost in the first $[N-1]$ epochs is lower bounded by

$$\sum_{\tau=1}^{N-1} \sum_{j=l_\tau+1}^{l_{\tau+1}} \|\mathbf{w}_j - \mathbf{w}_{j-1}\| > (N-1)\frac{c}{S}.$$

On the other hand, since the overall budget is $S$, we know

$$\sum_{\tau=1}^{N-1} \sum_{j=l_\tau+1}^{l_{\tau+1}} \|\mathbf{w}_j - \mathbf{w}_{j-1}\| \leq S.$$

Thus

$$N \leq \frac{S^2}{c} + 1. \tag{28}$$

Let $L_\tau$ be the length of epoch $\tau \in [N]$. Based on the Step 9, we know that for each epoch $\tau$, $\mathbf{m}_{l_\tau} M_{l_\tau-1} \geq 0$. Thus, we have

$$\left\| \sum_{t=1}^{T} \mathbf{m}_t \right\|^2 = \left\| L_N \mathbf{m}_{l_N} + \sum_{\tau=1}^{N-1} L_\tau \mathbf{m}_{l_\tau} \right\|^2 \geq \|L_N \mathbf{m}_{l_N}\|^2 + \left\| \sum_{\tau=1}^{N-1} L_\tau \mathbf{m}_{l_\tau} \right\|^2 \geq G^2 \sum_{\tau=1}^{N} L_\tau^2. \tag{29}$$

Thus

$$\left\| \sum_{t=1}^{T} \mathbf{m}_t \right\| \geq G \sqrt{\sum_{\tau=1}^{N} L_\tau^2} \geq G \frac{T}{\sqrt{N}} \geq G \frac{T\sqrt{c}}{\sqrt{S^2 + c}}, \tag{30}$$

where the first inequality is derived from (29), the second inequality is based on Cauchy-Schwarz inequality, and the final inequality is derived from (28).