# OpenReview forum: "Online Convex Optimization with Continuous Switching Constraint"
_NeurIPS.cc/2021/Conference — NeurIPS 2021 Poster_

### Official Review · Reviewer_2rzB · 2021-06-27

**Rating:** 7
**Confidence:** 2

**Summary:**

This paper studies online convex optimization with continuous switching constraints. The objective of the online agent is to minimize the static regret while keeping the total switching cost within some budget $S$. Under some standard assumptions, the authors establish a lower bound of the regret for all online algorithms by designing the adversary’s policy. They also show that the classical online gradient descent algorithm can match this lower bound if the step sizes are chosen appropriately. In addition, under stronger assumptions on the loss functions, the authors show the regret can be further improved to $O(\log{T})$.

**Limitations And Societal Impact:**

Yes, the authors discussed some limitations in the future work section. As a suggestion, I think it is also interesting to study the lower bounds for strongly convex loss functions.

**Main Review:**

This paper makes novel contributions to the online optimization literature. Although the proof techniques (e.g. linear games) and the OGD algorithm used in this paper is based on previous works, the authors make considerable changes to deal with the switching constraints. They compare their approach with previous works in details, and the intuitions behind these changes are discussed carefully.

I go through all the claims and proofs in the main body. They are easy to follow and look technically sound to me. There are a few typos that do not affect the correctness. In line 7 in the abstract, the bound should be $\Omega(\min\{T/S, T\})$; In the inequality after line 193, $\sum_{i=1}^{N-1}$ should be changed to $\sum_{\tau=1}^{N-1}$, and $L_i^2$ should be $L_\tau^2$.

This paper is well organized and should be easy to follow for an expert reader.

To the best of my knowledge, the results presented in this paper are interesting and significant. The problem setting is a natural variant of smoothed OCO and switching-constrained OCO. The bounds are technically strong because they establish the optimality in the proposed setting. There is also considerable originality in the proof of the lower bound result. It is possible that I overestimate the significance of this work due to lack of familiarity. Thus, I choose a low confidence level.


**Time Spent Reviewing:**

4

---

> ### Author Response · Authors · 2021-08-08
> **Many thanks for the constructive reviews!**
>
> Thank you very much for the constructive comments! We will fix the typos in the revised version.

---

### Official Review · Reviewer_fBbS · 2021-07-13

**Rating:** 6
**Confidence:** 5

**Summary:**

This work studies the well-known Online Convex Optimization (OCO) problem under an additional $\ell_2$-norm hard switching constraint of $\sum_{t=2}^T |w_t - w_{t-1}| \leq S$ where $w_t$'s are the actions of the algorithm and $S$ is the switching budget. They first provide hardness results to show that the regimes $S=\Omega (\sqrt{T})$ and $S=O(\sqrt{T})$ are different and have varying regret lower bounds. Then, they show that Online Gradient Descent (OGD) with particular choices of step size is able to obtain matching regret upper bounds in both regimes. Finally, the analysis is extended to the strongly convex setting.

**Limitations And Societal Impact:**

This is a theoretical work without any potential ethical concerns. The authors have done a good job describing the scope of their work and its limitations throughout the paper.


**Main Review:**

Strengths of the paper:
- The authors have done a great job motivating the use of continuous switching constraint (instead of a constraint on the number of switches which is used in prior works) through providing real-life examples where the former makes more sense.
- The related work section provides a comprehensive overview of all the prior works that share similar goals or techniques with this work and highlights the particular gap that this paper is going to cover.
- Algorithm 1 (for proving the lower bounds) is described in an intuitive way and it is easy to follow the arguments and the proof.

Drawbacks of the paper:
- The paper lacks any sort of numerical examples (even on synthetic data) to highlight the two different regimes for this problem and their different regret bounds. In particular, a plot showing the trade-off between the choice of $S$ and the regret would be really interesting.
- The smoothed OCO framework (and the Metrical Task System (MTS) problem in the competitive analysis setting) could be interpreted as the penalized formulation of OCO with continuous switching constraint. Therefore, the claim that these two settings are significantly different is not reasonable and it might be possible to draw some ideas from that literature to address this problem.
- If we model the OCO with continuous switching constraint as an OCO with long term constraints problem, the constraints would actually be static because at round $t\in[T]$, the constraint function is $|x-x_{t-1}|$ and so, the algorithm knows the exact form of the constraint before committing to action $x_t$. In this setting, the impossibility result of Mannor et al. does not apply and so, it might be possible to obtain meaningful results using this formulation.
- The mini-batch OGD algorithm of Chen et al. has been mentioned several times throughout the paper without a sufficient description of the algorithm. So, for readers that are not familiar with the prior works, it is hard to follow the arguments regarding the shortcomings of that algorithm.
- The details of the proofs (on pages 8 and 9) are not really insightful and most of them could have been moved to the appendix to provide enough space for numerical experiments in the paper.
**********************
Post-Rebuttal Update: Thanks to the authors for their detailed responses to the raised questions. With regards to OCO with long-term budget constraints, I'd like to note that given that this problem has two objectives (minimizing regret and constraint violation), it is always possible to minimize the constraint violation while incurring a larger regret and so, the notion of lower bound should be precisely defined. Also, prior works in this literature have provided Pareto-optimal curves for these two objectives.


**Time Spent Reviewing:**

2-3 hours

---

> ### Author Response · Authors · 2021-08-08
> **Many thanks for the constructive reviews!**
>
> Thank you very much for the constructive comments!
>
> ---
>
> Q1: The paper lacks any sort of numerical examples (even on synthetic data) to highlight the two different regimes for this problem and their different regret bounds. In particular, a plot showing the trade-off between the choice of $S$ and the regret would be really interesting.
>
> A1: Thank you for the advice. We would like to add empirical evaluations in the final version. Specifically, we will provide the cumulative regret v.s. iterations curves of our proposed algorithm and that of (Chen et al., 2020) under different choices of the budget.
>
> ---
>
> Q2: The smoothed OCO framework (and the Metrical Task System (MTS) problem in the competitive analysis setting) could be interpreted as the penalized formulation of OCO with continuous switching constraint. Therefore, the claim that these two settings are significantly different is not reasonable and it might be possible to draw some ideas from that literature to address this problem.
>
> A2: Thank you for the comments. We will revise our claims and add more discussions on this point. We agree that there is a strong connection between our proposed setting and smoothed OCO, and it’s definitely possible to draw ideas from that literature to apply to our problem. Nevertheless, we would like to note that, the smoothed OCO literature typically require more assumptions (i.e., observing $f_t$ before choosing $x_t$), and do not consider the trade-off between the choice of budget and the regret. By contrast, our setting exactly follows the classical OCO framework, and we mainly focus on studying the relationship between budget $S$ and the minimax regret.
>
> ---
>
> Q3: If we model the OCO with continuous switching constraint as an OCO with long term constraints problem, the constraints would actually be static because at round, the constraint function is $|x-x_{t-1}|$ and so, the algorithm knows the exact form of the constraint before committing to action. In this setting, the impossibility result of Mannor et al. does not apply and so, it might be possible to obtain meaningful results using this formulation.
>
> A3: Thank you for the comments. It is indeed interesting to investigate the relationship between OCO-CSC and OCO with long-term constraints. However, we would like to note that:
> 1. Although the player could observe $|x-x_{t-1}|$, the constraint at round $t$ is not static but still *time-variant* because $x_{t-1}$ depends on the algorithm used, and therefore most of the existing algorithms on OCO with long term constraints problem cannot be directly applied here.
> 2. As mentioned in our paper, we mainly focus on the studying the relationship between the budget $S$ and the regret, and our main contribution is the construction of the lower bound. However, to our knowledge, the lower bound for OCO with long term constraints problem (even for static constraints) is rarely studied.
>
> ---
>
> Q4: The mini-batch OGD algorithm of Chen et al. has been mentioned several times throughout the paper without a sufficient description of the algorithm. So, for readers that are not familiar with the prior works, it is hard to follow the arguments regarding the shortcomings of that algorithm.
>
> A4: Thank you for the advice. We will provide a detailed description of the mini-batch OGD algorithm in Section 2.2.
>
> ---
>
> Q5: The details of the proofs (on pages 8 and 9) are not really insightful and most of them could have been moved to the appendix to provide enough space for numerical experiments in the paper.
>
> A5: Thank you for the advice. We will move the proofs to the appendix, and add the experiments as described in Q1 and A1.

---

### Official Review · Reviewer_2Yte · 2021-07-13

**Rating:** 7
**Confidence:** 4

**Summary:**

The authors study online convex optimization with an L2 switching budget (S) on the actions chosen by the online algorithm. The authors propose a modification of the orthogonal technqiue of Abernathy et al to incorporate the switching constraint and propose a new lower bound on the regret achievable for this problem. The authors show a phase transition in the achievable regret for different values of S. The authors show that by setting the step size appropriately, vanilla OGD can match the lower bounds they present.

**Ethical Concerns:**

No problem here.

**Limitations And Societal Impact:**

No problem here.

**Main Review:**

This papers makes main two contributions. First, it proposes a modification of the orthogonal technqiue of Abernathy et al to incorporate the switching constraint, leading to a new lower bound which exhibits a phase transition as S varies. Both the phase transition result and the modification of the orthogonal technique are new and interesting. Second, they show that vanilla OGD with appropriate step size matches this lower bound exactly. This observation is surprising, given the simplicity of OGD and the fact that it does not explicitly consider the switching budget. I like these results.

I notice that in the related work section about smoothed OCO, the authors forget the paper "An online algorithm for smoothed regression and lqr control". I encourage the authors to add this citation.

The author's results all hold when the time horizon T is known in advance. It should be easy to convert their algorithm to an anytime algorithm using the doubling trick. This is a minor point, but the authors may wish to study this in future work. I also suspect that with a bit more work the authors could show that OGD matches the regret lower bounds all the way to the constant.

This paper shows new and interesting results for a problem which has attracted much recent attention. I vote accept.

**Time Spent Reviewing:**

1

---

> ### Author Response · Authors · 2021-08-08
> **Many thanks for the constructive reviews!**
>
> Thank you very much for the constructive comments!
>
> ---
>
> Q1: I notice that in the related work section about smoothed OCO, the authors forget the paper "An online algorithm for smoothed regression and lqr control". I encourage the authors to add this citation.
>
> A1: Thank you for the comments! We will add the missing citation in the revised version.
>
> ---
>
> Q2: it should be easy to convert their algorithm to an anytime algorithm using the doubling trick.
>
> A2: Thanks for the comments. We totally agree that our method could be extended to an anytime algorithm by making use of the doubling trick, and we will add more discussions on this point.
>
> ---
>
> A3: I also suspect that with a bit more work the authors could show that OGD matches the regret lower bounds all the way to the constant.
>
> Q3: Thanks for the comments. We will investigate this problem in the future.

---

### Official Review · Reviewer_uXtV · 2021-07-19

**Rating:** 5
**Confidence:** 4

**Summary:**

The paper investigates the problem of online convex optimization under metric switching constraints in the non-stochastic setting. In particular, the goal is to minimize a convex loss function where the long-term cumulative switching cost is subject to a stringent budget constraint. A lower bound which accounts for switching constraint is obtained. It is shown that gradient-descent achieves the minimax optimal regret bound.

**Limitations And Societal Impact:**

Yes.

**Main Review:**

The main contribution of this paper is to formulate the OCO problem with metric switching costs, and providing a lower bound for the regret. The main analysis tools seem to be straightforward, maybe except the adaptive adversary that bases its decisions on the history to achieve sharper lower bounds. The upper bound is achieved by the online gradient descent algorithm, which does not incorporate the switching cost dynamics but still achieves the minimax bound, thus the switching cost formulation does not yield a significant change in the algorithm behavior. This implies that the problem is somewhat simple, and limits the contribution of the paper.

- The proposed optimization algorithm is pretty straightforward, i.e., online gradient descent (OGD) with projection, therefore there is not much novelty in the algorithm design part. According to the proof of Theorem 2 in the Supplementary Material, the proof also follows from well-known analysis techniques.

- Interestingly, the only place where switching constraint $S$ comes into play is the step-size choice $\eta$ in OGD. Other than that, the algorithm ignores the switching constraint, and it still achieves minimax regret by a straightforward analysis. This implies that, technically, incorporating the switching cost constraints does not lead to a big challenge in the algorithm design.

- In case the switching constraint $S$ is not available a priori, could it be possible to find an adaptive policy to choose the step-size? Also, could it be possible to improve the regret coefficients by a different design which takes the history into account (similar to the lower bound)?

**Time Spent Reviewing:**

8

---

> ### Author Response · Authors · 2021-08-08
> **Many thanks for the constructive reviews!**
>
> Thank you very much for the constructive comments!
>
> ---
>
> Q1: The proposed optimization algorithm is pretty straightforward, i.e., online gradient descent (OGD) with projection, therefore there is not much novelty in the algorithm design part. According to the proof of Theorem 2 in the Supplementary Material, the proof also follows from well-known analysis techniques.
>
> A1: Thank you for the comments. We agree that our upper bound is straightforward to obtain. However, we would like to argue that *finding a simple (and optimal) solution to a well-motivated problem* should be regarded as an advantage, instead of a limitation. Moreover, we would like to emphasize the following points:
> 1. The main contribution of our paper, as agreed by all reviewers, is introducing the OCO-CSC problem as well as developing the lower bound. The upper bound, though simple, is minimax optimal, and perfectly matches the lower bound w.r.t. the main parameters $T$, $D$, and $G$.
> 2. Note that the algorithm design becomes straightforward only if one figures out what the lower bound is. Without the lower bound, it is unclear what the optimal bound looks like. Therefore, it is also not easy to see, at least for us, why we should use OGD with a constant step size instead of other methods, and why we should set the step size differently for different $S$ as in Eq (11).
>
> ---
>
> Q2: Interestingly, the only place where switching constraint $S$ comes into play is the step-size choice $\eta$ in OGD. Other than that, the algorithm ignores the switching constraint, and it still achieves minimax regret by a straightforward analysis.
>
> A2: Thanks for the comments. We agree the main idea of our proposed algorithm is to set different step sizes $\eta$ for different budgets $S$. However, we note that, tuning the step size of OGD, though simple, is very effective, and widely used in the classical OCO literature, such as OGD for strongly convex functions (Hazan et al, 2007), Aadagrad [1] or minibatch OGD (Chen et al., 2020).
>
> [1] Duchi et al., Adaptive Subgradient Methods for Online Learning and Stochastic Optimization. JMLR, 2011.
>
> ---
>
> Q3: This implies that, technically, incorporating the switching cost constraints does not lead to a big challenge in the algorithm design.
>
> A3: We agree with the reviewer, and this is indeed one implication of our theoretical results. We would also like to argue that, it might be non-trivial even if one is proving “incorporating a new thing does not lead to a big challenge in the algorithm design”, which  depends on whether the new thing as well as the minimax analysis are novel.
>
> ---
>
> Q4: In case the switching constraint is not available a priori, could it be possible to find an adaptive policy to choose the step-size?
>
> A4: Thank you for the question. It seems impossible because $S$ is the budget specified by the user. If $S$ is unknown, how can we distinguish between the traditional OCO and OCO-CSC? And how can we know whether the online algorithm exceeds the budget?
>
> To make the problem meaningful, we think the algorithm should have some prior knowledge about $S$ or can receive certain feedback from an Oracle. One interesting direction is to relax the setting and let the algorithm to restart once the condition is violated, and consider how to obtain a sublinear regret while minimizing the number of restarts. We leave it to the future work.
>
> ---
>
> Q5: Also, could it be possible to improve the regret coefficients by a different design which takes the history into account (similar to the lower bound)?
>
> A5: Thank you for the question. It is indeed possible to improve the constant factors with a more subtle analysis, and we will study this in the future work. We note that, currently, the upper bound is already minimax optimal w.r.t. the main parameters $T$, $D$, and $G$.

---

> ### Author Response · Authors · 2021-08-25
> **To Reviewer uXtV**
>
> Dear reviewer,
>
> Please let us know if your concerns are addressed by our rebuttal. We would be glad to answer any further questions that you have.
>
> Thank you!\
> Authors

---

### Decision · Program_Chairs · 2021-09-27

**Decision:**

Accept (Poster)

**Comment:**

This paper studies an interesting variant of OCO with continuous switching constraints. I have found the contributions of the paper quite interesting and novel. This is a very nice extension to the work by Chen et al 2020. So I recommend acceptance.